# Predictors and incidence of depression and anxiety in women undergoing infertility treatment: A cross-sectional study

**Lingyan Wang** [1], **Youyin Tang**[2], **Yuyang Wang**[1]*

**1** Department of Obstetrics and Gynecology Nursing, Key Laboratory of Birth Defects and Related Disease of Women and Children, West China Second University Hospital, Sichuan University, Chengdu P.R. China, **2** Department of Vascular Surgery, West China Hospital, Sichuan University, Chengdu, P.R. China

☯ These authors contributed equally to this work.
* 1536803465@qq.com

**Data Availability Statement:** All relevant data are within the paper and its Supporting Information files.

## Abstract

The global incidence of infertility is increasing year by year, and the association between infertility and mental illness has been widely concerned. The aim of this study was to investigate the incidence of anxiety and depression in infertile women in China and explore the risk factors which might lead to anxiety and depression. From January 2020 to December 2020, female infertile patients who received assisted reproduction technology (ART) treatment at West China Second Hospital were recruited and a total of 1712 eligible female patients were finally enrolled in this study. Generalized Anxiety Disorder-7 (GAD-7) and Patient Health Questionaire-9 (PHQ-9) were used to evaluate the patients' psychological status. The reliability of all scales was evaluated by Cronbach's α and Spearman-Brown half coefficient, and Kaiser-Meyer-Olkin (KMO) value was calculated by factor analysis to evaluate validity. Univariate and multivariate logistic regression analysis were applied for assessing independent risk factors of anxiety and depression, respectively. The incidence of anxiety and depression in infertile women were 25.2% and 31.3%, respectively. Cronbach's α coefficients and Spearman-Brown half-fold coefficients of GAD-7 and PHQ-9 were 0.870, 0.825, 0.852 and 0.793, respectively. Univariate and multivariate logistic regression analysis showed that education level (junior college degree or above, OR:1. 6, 95% CI: 1.2–2.1, P = 0.003), somatic symptoms (severe somatic symptoms, OR:15.2, 95%CI: 5.6–41.3, P<0.001), sleep quality (poor sleep quality, OR:9.3, 95% CI:4.7–18.4, P<0.001) were independent risk factors for anxiety. And age>35 years old, moderate and severe somatic symptoms and poor sleep quality were independent risk factors for depression. Somatic symptoms and poor sleep quality are both the risk factors of anxiety and depression symptoms of infertile woman. And high educated (junior college degree or above) patients are more likely to be complicated with anxiety symptoms, while elderly patients (age>35) are prone to be complicated with depression symptoms.

**Funding:** The author(s) received no specific funding for this work.

**Competing interests:** The authors have declared that no competing interests exist.

## Introduction

Infertility is a disease of reproductive system, which is defined as the failure to conceive despite 12 months of unprotected intercourse [1]. Studies have revealed that environmental pollution, food safety, alcoholism, smoking, endocrine dysfunction, abnormal immune function, delayed childbearing age and other factors will contribute to an increase risk of couples suffered from infertility [2]. There are nearly 200 million people diagnosed with infertility worldwide [3]. The incidence of infertility is increasing year by year in China [4]. Previous studies have shown that there were more than 50 million infertile patients in China in 2014, with a prevalence rate of 15% [5]. However, by 2023, the prevalence rate of infertility is expected to be as high as 18.2% [4, 6, 7].

Previous studies have shown that besides being unable to conceive naturally, infertility is also related to various physical and mental diseases such as anxiety, depression, obsessive-compulsive symptoms and sleep disorder [8, 9]. In the past years, infertility was considered as a shameful disease, and society and family might give individuals and couples a greater psychological burden which greatly affected the physical and mental health of infertile patients [10]. For infertile patients, first went to outpatient consultation or chose assisted reproductive technology treatment might cause anxiety [11]. In Japan, about 67% of unmarried women believe that couples should have at least one child after marriage, while about 25% of married women believe that a couple is socially acceptable only after they have children, according to the study [12]. In China, people attach great importance to fertility [13, 14]. Even though women have relatively high family decision-making power than in the past, there still exist a tendency in society to think that "it is always women's fault to have no children" [5, 15]. Therefore, compared to male patients, female infertility patients experienced greater psychosocial stress and were more likely to induce psychosocial related physical and mental disorders, such as anxiety, depression, obsessive compulsive symptoms, and sleep disorders [16].

In recent years, with the increasing incidence of infertility, more and more researchers have begun to pay attention to the psychological stress, such as anxiety and depression, of infertile women [5, 9, 17]. However, many of the current studies only focused on the relationship between stigma and psychological distress [4, 5, 12, 18], only few studies [13, 14] investigated the risk factors for psychological stress in infertile women. Besides, the sample size included in these studies were not large enough, and they only investigated the influence of demographic characteristics on psychological stress of infertile women. Thus, a large sample size study with more variables focused on psychological stress of infertile female patients was helpful for clinicians to accurately predict the high-risk factors and severity of psychological stress of infertile female patients and guide clinical decision-making.

Based on the above reasons, we designed a cross-sectional study to investigate the incidence of anxiety and depression symptoms of infertile female patients in West China Second Hospital of Sichuan University, aiming at determining the risk factors causing anxiety and depression symptoms and predicting the risk of anxiety and depression of infertile female patients.

## Materials and methods

### Study design and setting

The demographic characteristics, somatic symptoms, sleep quality and psychological stress of female infertile patients in our hospital were collected by questionnaire. Psychological stress refers to anxiety and depression. Patient health questionnaire-15 (PHQ-15) was used to record physical symptoms and Pittsburgh sleep quality index (PSQI) was used to record sleep quality. Anxiety and depression were assessed with Generalized Anxiety Disorder (GAD-7) and

Patient Health Questionaire-9 (PHQ-9), respectively. In addition, anxiety and depression symptoms (converted into binary variables) were taken as dependent variables, and physical symptoms, sleep quality and demographic characteristics were taken as independent variables. Univariate and multivariate logistic regression analysis was carried out to select independent risk factors and quickly identify high-risk patients, so as to facilitate early psychological intervention.

### Patient selection

During January 2020 to December 2020, female infertile patients who received assisted reproduction technology (ART) treatment at infertility counseling department of West China Second Hospital were recruited and medical records of them were reviewed, and a total of 1712 female infertile patients were included in this study.

The inclusion criteria were: i) patient should be older than 20 years old; ii) patient had received primary school education and above and able to communicate normally; iii) patient had good physical condition and had not been diagnosed with mental illness before. Exclusion criteria: i) patient had impairment of reading, comprehension or writing; ii) patient was diagnosed with consciousness disorder and cognitive disorder of orientation.

### Questionnaire design and content

Demographic characteristics include age, educational, occupation, causes of infertility. Physical symptoms (evaluated by PHQ-15) include 15 items such as abdominal pain, back pain, arm, leg or joint pain, dysmenorrhea, sexual intercourse pain, etc., and sleep quality (evaluated by PSQI) include 7 items (sleeping time, sleep efficiency, hypnotic drugs, etc.). The scores of GAD-7, PHQ-9, PHQ-15 and PSQI [19] in patients were shown in **S1 File**. The questionnaires were in Chinese format. A nurse (Lingyan Wang) with a National Psychological Consultant certification distributed and retrieved the questionnaires. And all the questionnaires were finished in the infertility counseling department of West China Second Hospital during January 2020 to December 2020. All the questionnaires were locked in a special box that could only be opened by Yuyang Wang. A monthly questionnaire count was performed independently by Lingyan Wang, Youyin Tang, and Yuyang Wang.

### Generalized anxiety disorder-7 (GAD-7) [20]

The scale for evaluating patients' generalized anxiety symptoms and severity, one month prior to the study consists of 7 scoring items, and each item is scored according to 4 grades (0, 1, 2 and 3), with a total score ranging from 0 to 21. The higher the score, the more serious the anxiety symptoms. Grading criteria: 0 to 4 for no anxiety, 5 to 9 for mild anxiety, 10 to 14 for moderate anxiety and 15 to 21 for severe anxiety.

### Patient health questionnaire-9(PHQ-9) [21]

To evaluate the psychological condition of patients in the past 2 weeks, there were 9 items in total, and each item was scored according to 4 grades (0, 1, 2 and 3), with a total score ranging from 0 to 27. The higher the score, the more severe the depressive symptoms. Grading criteria: 0 to 4 for no depression, 5 to 9 for mild depression, 10 to 14 for moderate depression and 15 to 27 for severe depression.

### Quality control and reliability & validity analysis [22]

The two researchers who participated in the quality control of this study were trained professionals (with the national third-class and above psychological counselor qualification certificate) to ensure the authenticity and validity of the questionnaire and scale. In this study, patients do not need to fill in private information such as their names and ID numbers, and all patients voluntarily participate in this study and sign informed consent forms. At the same time, researchers are not allowed to guide, mislead or fill in on their behalf.

Cronbach's α coefficient and Spearman-Brown half-fold coefficient were used to analyze the reliability of the scale. Generally, Cronbach's α coefficient> 0.7 and Spearman-Brown half-fold coefficient> 0.7 suggesting that the internal consistency of the questionnaire is good, that is, the reliability is good. Factor analysis was used to evaluate the structure validity of each scale by calculating the KMO value. KMO value> 0.6 indicates a well data structure validity. Calibration validity refers to the proximity between the scale measurement results and the standard measurement.

### Ethical considerations

This study was a cross-sectional study which was approved by Medical Ethics Committee of West China Second Hospital of Sichuan University (Ethic number: No.2020167). Written informed consent was waived due to no individual information (such as patient's name, ID card number, phone number and biometrics identification data) was identified.

### Statistical analysis

Continuous variables are expressed by mean ± standard deviation (x± s), and t test is used for statistical analysis. Counting data is expressed as number (n), and chi-square test is applied for statistical analysis. The reliability of all scales was evaluated by Cronbach's α and Spearman-Brown half coefficient, and KMO value was calculated by factor analysis to evaluate validity. Univariate and multivariate Logistic analysis was used to analyze the independent risk factors leading to anxiety symptoms or depression symptoms. Variables with p-value<0.1 in univariate analysis were further stepped in multivariate Logistic regression analysis. The test level α was 0.05. P value< 0.05 was considered statistically significant. All statistical analyses were conducted with EmpowerStats software, version 2.20 and SPSS20.0 software.

## Results

### Demographic characteristics of patients

In this study, a total of 2301 female infertility patients were given questionnaires, among which 438 patients refused to participate in the survey, and 151 patients presented incomplete questionnaires. The flow diagram of this study was showed in (Fig 1). A total of 1712 valid questionnaires were recovered, with a questionnaire recovery rate of 74.4%. The patients ranged from 20 to 49 years old, with an average age of 32.1±4.59 years old. Besides, the incidence of anxiety and depression in infertile women were 25.2% and 31.3%, respectively. Among all the patients included in this study, there was 836 patients whose age range from 30 to 35 years old with the highest composition ratio of 48.8%. While 1118 patients had an education level of junior college or above, accounting for 65.3% of all patients. The differences were statistically significant (P< 0.05) (Table 1).

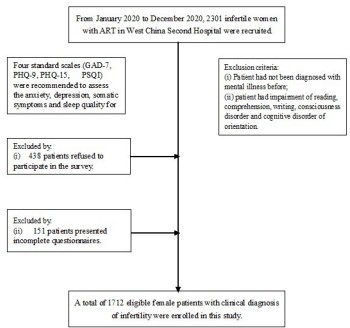

**Fig 1. The flow diagram of this study.** Abbreviation: GAD-7: Generalized Anxiety Disorder 7; PHQ-15: Patient health questionnaire-15; PSQI: Pittsburgh sleep quality index; PHQ-9: Patient Health Questionaire-9.

**Table 1. The baseline characteristics of infertility women in this study.**

| Variable | Anxiety symptoms(assessed by GAD-7) | | | |
|---|---|---|---|---|
| | No anxiety (n = 1267) | Mild anxiety (n = 383) | Moderate anxiety (n = 49) | Severe anxiety (n = 13) |
| **Age, year, n** | | | | |
| (20, 29] | 428 | 132 | 12 | 5 |
| (30, 35] | 601* | 199 | 30 | 6 |
| >35 | 238 | 52 | 7 | 2 |
| **Educational level, n** | | | | |
| Junior high school | 247 | 49 | 7 | 4 |
| High school | 220 | 59 | 8 | 0 |
| Junior college degrees or above | 800* | 275 | 34 | 9 |
| **Occupational status, n** | | | | |
| Employed | 603 | 208 | 27 | 6 |
| Un-employed | 664 | 175 | 22 | 7 |
| **Causes of infertility, n** | | | | |
| Primary infertility | 557 | 184 | 23 | 5 |
| Secondary infertility | 710 | 199 | 26 | 8 |
| **Sleep quality#, n** | | | | |
| Very well | 819 | 91 | 2 | 1 |
| General | 431 | 257 | 33 | 4 |
| Poor | 17 | 34 | 14 | 8 |
| Awful sleep quality | 0 | 1 | 0 | 0 |
| **Somatic Symptoms&, n** | | | | |
| No | 924 | 107 | 6 | 1 |
| Mild | 302 | 170 | 18 | 3 |
| Moderate | 36 | 92 | 18 | 5 |
| Severe | 5 | 14 | 7 | 4 |

**Abbreviation:** GAD-7: Generalized Anxiety Disorder 7

\# Assessed by Pittsburgh sleep quality index (PSQI)

&Assessed by Physical Health Questionnaire-15(PHQ-15).

\* Statistically difference (P<0.05)

### Reliability, construction validity and spearman correlation analysis of the scale

Cronbach's α coefficients of PHQ-15, GAD-7, PHQ-9 and PSQI were 0.809, 0.870, 0.825 and 0.758 respectively. Spearman-Brown half-fold coefficients were 0.700, 0.852, 0.793 and 0.783 respectively, and KMO values were 0.892, 0.890, 0.913 and 0.862 respectively. Cronbach's α coefficients and Spearman-Brown half-fold coefficients suggested that the reliability and construction validity of the four scales included in this study are good (P < 0.001). **(Shown in S1 File)**.

Spearman correlation analysis showed that the correlation coefficients was well (P < 0.001). (**Shown in S1 File)**.

### Univariate and multivariate logistic regression analysis of anxiety and depressive symptoms

Univariate and multivariate logistic analysis showed that education level (junior college degree or above, OR:1. 6, 95%CI: 1.2–2.1, P = 0.003), somatic symptoms (severe somatic symptoms, OR:15.2, 95%CI: 5.6–41.3, P<0.001), sleep quality (poor sleep quality, OR:9.3, 95%CI:4.7–18.4, P<0.001) were independent risk factors for anxiety. The univariate analysis and multivariate logistic analysis of anxiety was showed in **Table 2**.

Univariate and multivariate Logistic analysis showed that age>35 years old, moderate and severe somatic symptoms and poor sleep quality were independent risk factors for depression. Severe somatic symptoms and poor sleep quality had a very strong contribution to anxiety, with an Odds Ratio of 13.3 and 16.1 respectively. The univariate analysis and multivariate Logistic analysis of depress was showed in **Table 3.**

## Discussion

We conducted a cross-sectional study to investigate the prevalence of anxiety and depression among women treated for infertility and the independent risk factors for anxiety and depression. This study found that anxiety symptoms and depression in infertile women were significantly correlated with their educational level, physical symptoms and sleep quality, rather than occupation status and infertility reasons. In addition, age> 35 years old is an independent risk factor for depression. Therefore, psychoeducation is necessary for infertile women who combined the above risk factors to reduce the risk of anxiety or depression. The main findings and impacts are discussed as follows.

In China, people attach great importance to fertility, and there is a tendency in society to think that "it is always a woman's fault to have no children" [5]. This study found that the incidence of anxiety and depression in infertile women were 25.2% and 31.3%, respectively, which is lower than the incidence of anxiety and depressed reported in Japan of 51.1% and 54%, respectively [12]. It might be due to Japan's developed economy and high social pressure that young women with infertility were more likely to experience anxiety and depression.

Multivariate logistic regression analysis found that moderate and severe somatic symptoms were independent risk factors of anxiety (P<0.001) and depression in infertile patients (P< 0.001), and severe somatic symptoms can increase the risk of anxiety symptoms by 14.2-fold. Further analysis of the sources of patients' physical symptoms shows that dysmenorrhea and other menstrual problems account for 58% (P<0.05). In addition, many studies have shown that all kinds of pain can increase the risk of anxiety and depression, which were consistent with the conclusion of our study [23–25]. Similarly, a study of women with dysmenorrhea found that they were also more likely to experience anxiety and depression [26]. Therefore,

**Table 2. Univariate analysis and multivariate logistic regression analysis of the anxiety in infertile women.**

| Variable | Univariate analysis (*p* value) | Multivariate analysis OR (95%CI) | *p* value |
|---|---|---|---|
| **Age, year** | | | |
| (20, 29] | ref | ref | ref |
| (30, 35] | 0.28 | 1.1(0.8, 1.4) | 0.52 |
| >35 | 0.06 | 0.8(0.5, 1.2) | 0.22 |
| **Occupational status** | | | |
| Employed | ref | ref | ref |
| Un-employed | 0.71 | 1.2(0.8, 1.8) | 0.77 |
| **Educational background** | | | |
| Junior high school | ref | ref | ref |
| Higher school | 0.27 | 1.4(0.9, 2.2) | 0.13 |
| Junior college degrees or above | <0.001 | 1.6(1.2, 2.1) | 0.003 |
| **Somatic Symptoms*** | | | |
| No | ref | ref | ref |
| Mild | <0.001 | 3.4(2.6, 4.5) | <0.001 |
| Moderate | <0.001 | 13.3(9.3, 19.1) | <0.001 |
| Severe | <0.001 | 15.2(5.6, 41.3) | <0.001 |
| **Causes of infertility** | | | |
| Primary infertility | ref | ref | ref |
| Secondary infertility | 0.88 | 1.0(0.8, 1.3) | 0.26 |
| **Sleep quality#, n** | | | |
| Very well | ref | ref | ref |
| General | <0.001 | 3.3(2.6, 4.2) | <0.001 |
| Poor | <0.001 | 9.3(4.7, 18.4) | <0.001 |
| Awful sleep quality | 0.97 | - | - |

**Abbreviation**: ref: reference; OR: odds ratio; CI: confidence interval

*Assessed by Physical Health Questionnaire-15(PHQ-15).

# Assessed by Pittsburgh sleep quality index (PSQI)

clinical attention should be paid to patients' dysmenorrhea and other somatic symptoms in order to reduce their risk of anxiety and depression.

Infertile women were 16.1 times more likely to be depressed when their sleep quality is poor, showing a strong connection between poor sleep quality and depression. Previous investigations [27–29] implied that insomnia can increase the risk of depression or anxiety through impair emotional regulation, which supported our result. In addition, analysis of the PSQI scale revealed that 71.8% of patients suffered from subjective sleep disorders, but only 0.99% of patients used hypnotic drugs for sleep. Although we don't know why only a few patients use hypnotics to sleep, we think that the sleep quality of patients can be improved by strengthening the preaching of safe use of hypnotics to help patients sleep and guiding patients to actively go to sleep medical centers.

The study also found that having a junior college degree or higher increased the risk of anxiety symptoms (P = 0.003), but not depression symptoms (P = 0.089). Although the impact of education level on anxiety is controversial, the majorities believed that higher education level will increase the incidence of anxiety disorder, especially for women, this is because the current labor market cannot provide adequate economic or social resources for people with higher education level, and this mismatch between labor and salary will lead to depression and

Table 3. Univariate analysis and multivariate logistic regression analysis of the depression in infertile women.

| Variable | Univariate analysis (*p* value) | Multivariate analysis OR (95%CI) | *p* value |
|---|---|---|---|
| **Age, year** | | | |
| (20, 29] | Ref | ref | Ref |
| (30, 35] | 0.670 | 1.0(0.8, 1.3) | 0.99 |
| >35 | 0.012 | 0.7(0.5, 0.98) | 0.044 |
| **Occupational status** | | | |
| Employed | Ref | ref | ref |
| Un-employed | 0.167 | 0.9(0.7, 1.3) | 0.90 |
| **Educational background** | | | |
| Junior high school | Ref | ref | ref |
| Higher school | 0.769 | 1.0(0.6, 1.5) | 0.88 |
| Junior college degrees or above | 0.016 | 1.3(1.0, 1.8) | 0.089 |
| **Physical Symptoms*** | | | |
| No | ref | ref | ref |
| Mild | <0.001 | 4.0(3.1, 5.2) | <0.001 |
| Moderate | <0.001 | 14.8(8.9, 24.6) | <0.001 |
| Severe | <0.001 | 13.3(4.0, 44.2) | <0.001 |
| **Causes of infertility** | | | |
| Primary infertility | Ref | ref | ref |
| Secondary infertility | 0.736 | 0.7(0.5, 0.9) | 0.99 |
| **Sleep quality#, n** | | | |
| Very well | Ref | ref | ref |
| General | <0.001 | 3.1(2.6, 3.7) | <0.001 |
| Poor | <0.001 | 16.1(7.1, 36.5) | <0.001 |
| Awful sleep quality | 0.965 | - | - |

**Abbreviation**: ref: reference; OR: odds ratio; CI: confidence interval

*Assessed by Physical Health Questionnaire-15(PHQ-15).

# Assessed by Pittsburgh sleep quality index (PSQI)

disappointment of people with higher education, which might further result in mental illness [30]. In China, with the development of economy, women usually receive good education, however, the ensuing saturation of labor market will also increase the life and work pressure of higher education people, and then bring psychological stress. Previous studies have suggested that the influence of education level on mental health is not simply increasing linearly, but there is a threshold. Once the education level is higher than this threshold, the risk of mental illness will greatly increase [31, 32], which is also consistent with the current conclusion of our research that highly educated infertile patients are more likely to have anxiety symptoms. Although we didn't deeply explore why high education level has such a different impact on anxiety and depression, it is completely consistent with the relationship between Japanese infertile women's education level and anxiety or depression reported in Yokota's research [12], which further proves the reliability of our research conclusions. Therefore, we should provide the psychological test and offer psychoeducation for highly educated infertile women to reduce the impact of anxiety and depression on them.

The age of infertile woman (>35 years) was another independent risk factor for depressive symptoms (P<0.05). These finding was similar to many previous study conducted in other provinces of China [13, 14]. And the possible reason might be that with the increase of age, the probability of successful conception of infertile patients will gradually decrease [33]. A large

clinical study involving 1612 women with a total of 4246 insemination cycles showed that when the age of infertile patients is over 35 years old, the clinical pregnancy rate after intra-uterine insemination will decrease from 27.1% to 21.6% [34], which may be the potential cause of depression in elderly patients. Consequently, infertile women who over 35 might benefit from psychoeducation to reduce anxiety and depression.

However, there are still some limitation in this study. Firstly, this study is a single-center cross-sectional study, and there is inevitable selectivity bias. The results of this study still need to be confirmed by multicenter prospective studies. In addition, the high rate of lost follow-up among infertile outpatients makes it difficult to reevaluate all patients, which will have an impact on the study's repeatability. Finally, during the study period, COVID-19 pandemic broke out, and the psychological state of infertile women patients may be affected by COVID-19 pandemic, so it might have an impact on the research conclusion. However, most of the patients included in the study were from Sichuan Province which the epidemic is not serious in this area. Thus, the selection bias of this study in COVID-19 pandemic was relatively low. And we will carry out a large-scale multicenter related research in order to draw more scientific conclusions and help clinicians make more comprehensive clinical decisions in the future.

## Conclusions

Moderate and severe somatic symptoms and poor sleep quality can significantly increase the risk of anxiety symptoms and depression symptoms of infertile woman. And high educated (junior college degree or above) patients are more likely to be complicated with anxiety symptoms, while elderly patients (age>35) are more likely to be complicated with depression symptoms. Therefore, psychoeducation is necessary for infertile women who combined the above risk factors to reduce the risk of anxiety or depression.

## Supporting information

**S1 File. The details of four scales and reliability and validity analysis of four scales.** (DOCX)

## Acknowledgments

Thanks to the research platform provided by West China Second Hospital of Sichuan University and the statistical assistance provided by West China Big Data Center.

## Author Contributions

**Conceptualization:** Lingyan Wang, Youyin Tang, Yuyang Wang.

**Data curation:** Lingyan Wang, Youyin Tang.

**Formal analysis:** Lingyan Wang, Youyin Tang.

**Investigation:** Lingyan Wang, Youyin Tang, Yuyang Wang.

**Methodology:** Lingyan Wang, Youyin Tang.

**Project administration:** Lingyan Wang, Youyin Tang.

**Resources:** Lingyan Wang, Youyin Tang.

**Software:** Lingyan Wang, Youyin Tang.

**Supervision:** Lingyan Wang, Youyin Tang, Yuyang Wang.

**Validation:** Lingyan Wang, Youyin Tang, Yuyang Wang.

**Visualization:** Lingyan Wang, Youyin Tang, Yuyang Wang.

**Writing – original draft:** Lingyan Wang, Youyin Tang.

**Writing – review & editing:** Yuyang Wang.

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
