## [Decision Letter · Decision Letter 0]

20 Mar 2023

PONE-D-23-04196Predictors of depressive and anxiety in women undergoing infertility treatment: a cross-sectional studyPLOS ONE

Dear Dr. Wang,

Thank you for submitting your manuscript to PLOS ONE. After careful consideration, we feel that it has merit but does not fully meet PLOS ONE’s publication criteria as it currently stands. Therefore, we invite you to submit a revised version of the manuscript that addresses the points raised during the review process.

We look forward to receiving your revised manuscript.

Kind regards,

Jacopo Di Giuseppe

Academic Editor

PLOS ONE

Journal Requirements:

2. Please include your tables as part of your main manuscript and remove the individual files. Please note that supplementary tables (should remain/ be uploaded) as separate "supporting information" files

Reviewers' comments:

Reviewer's Responses to Questions

**Comments to the Author**

1. Is the manuscript technically sound, and do the data support the conclusions?

Reviewer #1: Partly

Reviewer #2: Yes

Reviewer #3: Yes

2. Has the statistical analysis been performed appropriately and rigorously? 

Reviewer #1: I Don't Know

Reviewer #2: Yes

Reviewer #3: Yes

3. Have the authors made all data underlying the findings in their manuscript fully available?

Reviewer #1: Yes

Reviewer #2: No

Reviewer #3: Yes

4. Is the manuscript presented in an intelligible fashion and written in standard English?

Reviewer #1: No

Reviewer #2: Yes

Reviewer #3: Yes

5. Review Comments to the Author

Reviewer #1: TITLE: Depressive should be substituted with depression and this should also be done in the short title

ABSTRACT: Lines 24-25 is not clear and should be recasted. In lines 28-29, CAD, PHQ, KMO should all be written in full 1st . The authors should include a summary of how the patients were selected and state the study design here. Lines 39-42, it is not clear if the authors are reporting results and there is not conclusion in the abstract.

INTRODUCTION: There are several grammatical errors and wrong use of tenses that make several statement difficult to understand. The statements in the following lines are not clear and should be recasted, 56-58, 58-59, 64-67, 70-72, 74-76, . Line 78 should be --- determining the risk factors---.

METHODS: Several grammatical errors and wrong use of tenses. How were the patients selected and how did the authors determine their sample size? The questionnaires should all be referenced. Line 112 should read --- severity , one month prior to the study---- . Who distributed the questionnaires, how were they administered , when and where was this done and what language were they? Under ethical considerations, it is not clear if approval for conduct of the study was given by the ethical review committe and in line 138 , what individual information are the authors referring to?

RESULTS: It is not clear in lines 159-160 which differences were statistically significant.

DISCUSSION: There is no need to repeat p values or odds ratios that are in the results here. There are also very many grammatical errors and wrong use of tenses that make many sentences difficult to understand. The following sentences should be recast6ed for clarity,; lines 212-213, 216-217, 219-220, 227-229, in 247 what do the authors mean by -- another founding--?. Lines 254-255 and 258-259 are not clear.

REFERNCES. The authors should crosscheck the journal title of reference number 2. The following references do not seem complete; 8,9,10,12,17 and 18

Reviewer #2: I suggest to include the incidence in the title because incidence was mentioned as one of the objectives.

Discussion

the rate (incidence of the anxiety was not discussed. Authors jumped directly to associated factors.

Incidence have to be compared with previous studies and to discuss who there is difference between the other studies.

I think sleep problem or pattern are not predictors for anxiety and depression but are effects. e,g are manifestation for anxiety and depression.

Reviewer #3: The article is an interesting one.

The methodology is not detailed enough. For instance, the administration of the questionnaire was not clearly described. I got to know about it in the result section.

The authors should do the corrections in the manuscript as suggested before publication.

6. PLOS authors have the option to publish the peer review history of their article (what does this mean?). If published, this will include your full peer review and any attached files.

Reviewer #1: No

Reviewer #2: No

Reviewer #3: **Yes: **Habiba Ibrahim Abdullahi

---

## [Author Response · Author response to Decision Letter 0]

25 Mar 2023

Reviewer #1:

(i) TITLE: Depressive should be substituted with depression and this should also be done in the short title

Thanks for your suggestion and we appreciated for your carefully review. We have changed this word.

(ii) ABSTRACT: Lines 24-25 is not clear and should be recasted. In lines 28-29, CAD, PHQ, KMO should all be written in full 1st. The authors should include a summary of how the patients were selected and state the study design here. Lines 39-42, it is not clear if the authors are reporting results and there is not conclusion in the abstract.

Thanks for your suggestions and we appreciated for your carefully review. We have recasted the abstract. And since “the format criteria of PLOs One” the conclusion can be found in the last two paragraph of abstract “Somatic symptoms and poor sleep quality are both the risk factors of anxiety and depression symptoms of infertile woman. And high educated (junior college degree or above) patients are more likely to be complicated with anxiety symptoms, while elderly patients (age>35) are prone to be complicated with depression symptoms.”

(iii) INTRODUCTION: There are several grammatical errors and wrong use of tenses that make several statement difficult to understand. The statements in the following lines are not clear and should be recasted, 56-58, 58-59, 64-67, 70-72, 74-76, . Line 78 should be --- determining the risk factors---.

Thanks for your questions and suggestions and we appreciated for your carefully review. We have revised our manuscript.

(iv) METHODS: Several grammatical errors and wrong use of tenses. How were the patients selected and how did the authors determine their sample size? The questionnaires should all be referenced. Line 112 should read --- severity, one month prior to the study----. Who distributed the questionnaires, how were they administered, when and where was this done and what language were they? Under ethical considerations, it is not clear if approval for conduct of the study was given by the ethical review committee and in line 138, what individual information are the authors referring to?

Thanks for your questions and suggestions and we appreciated for your carefully review. We have revised our manuscript.

For patient selection, all patients who came to our hospital for infertility counseling during January 2020 to December 2020 were asked to finish the questionnaires and the inclusion and exclusion criteria were shown in the manuscript. EmpowerStats software was used to calculate the sample size of the study. Power analysis of the study were shown in the following table. We chose the lowest hazards ratio (college degrees or above:1.6) to calculate the estimated samples size, and the estimated samples size should be more than 914.

Anxiety/No anxiety value Depression/ No depression value

Two-sided α 0.05 Two-sided α 0.05

The incidence of anxiety in infertile women 0.252 The incidence of depression in infertile women 0.056

Assuming hazard ratio of Junior college degrees or above 1.6 Assuming hazard ratio of general sleep quality 3.1

Power 90% Power 90%

Estimated patients number in this study 914 Estimated patients number in this study 136

The questionnaires were in Chinese format and distributed by a nurse who have obtained a certification of National Psychological Consultant, her name was Lingyan Wang. And all the questionnaires were finished in the infertility counseling department of West China Second Hospital. All the questionnaires were locked down in a special box, and counted every month by Lingyan Wang, Youyin Tang and Yuyang Wang.

Written informed consent was waived due to no individual information (such as patient’s name, ID card number, phone number and biometrics identification data) was identified.

(v) RESULTS: It is not clear in lines 159-160 which differences were statistically significant.

Thanks for your question and we appreciated for your carefully review. In this sentence, we mean to say that the age and education level had significant difference in no anxiety patients. And we add a “*” in table 1 to express this statistically difference.

(vi) DISCUSSION: There is no need to repeat p values or odds ratios that are in the results here. There are also very many grammatical errors and wrong use of tenses that make many sentences difficult to understand. The following sentences should be recasted for clarity; lines 212-213, 216-217, 219-220, 227-229, in 247 what do the authors mean by -- another founding--?. Lines 254-255 and 258-259 are not clear.

Thanks for your question and we appreciated for your carefully review. We have revised our manuscript according to the above questions. 

(vii) REFERNCES. The authors should crosscheck the journal title of reference number 2. The following references do not seem complete; 8,9,10,12,17 and 18

Thanks for your question and we appreciated for your carefully review. We have checked our references and revised some of them.

Reviewer #2:

(i) I suggest to include the incidence in the title because incidence was mentioned as one of the objectives.

Thanks for your suggestion and we appreciated for your carefully review. We have changed the title.

(ii) Discussion

the rate (incidence of the anxiety was not discussed. Authors jumped directly to associated factors. Incidence have to be compared with previous studies and to discuss who there is difference between the other studies.

Thanks for your suggestion and we appreciated for your carefully review. Actually, we have discussed the incidence of anxiety and depression and compared with previous studies in Line 225-227 in Discussion part. Besides, we added more details of the incidence and adjusted the position of this paragraph.

(iii) I think sleep problem or pattern are not predictors for anxiety and depression but are effects. e,g are manifestation for anxiety and depression.

Thanks for your question and we appreciated for your carefully review. We agree with your opinion that sleep problem might be manifestation for anxiety and depression. In this study we only want to explain the relationship between sleep problem and anxiety or depression, and we noticed that poor sleep quality patients had a strong association with depression. Therefore, we revised our manuscript according your suggestion.

Reviewer #3:

(i) The article is an interesting one. The methodology is not detailed enough. For instance, the administration of the questionnaire was not clearly described. I got to know about it in the result section. 

Thanks for your suggestions and we appreciated for your carefully review. We have revised our manuscript according your suggestion. The administration of the questionnaire were as follows.

“The questionnaires were in Chinese format. A nurse (Lingyan Wang) with a National Psychological Consultant certification distributed and retrieved the questionnaires. And all the questionnaires were finished in the infertility counseling department of West China Second Hospital during January 2020 to December 2020. All the questionnaires were locked in a special box that could only be opened by Yuyang Wang. A monthly questionnaire count was performed independently by Lingyan Wang, Youyin Tang, and Yuyang Wang.”

---

## [Editor Report · Decision Letter 1]

30 Mar 2023

Predictors and incidence of depression and anxiety in women undergoing infertility treatment: a cross-sectional study

PONE-D-23-04196R1

Dear Dr. Wang,

We’re pleased to inform you that your manuscript has been judged scientifically suitable for publication and will be formally accepted for publication once it meets all outstanding technical requirements.

Kind regards,

Jacopo Di Giuseppe

Academic Editor

PLOS ONE

---

## [Editor Report · Acceptance letter]

4 Apr 2023

PONE-D-23-04196R1 

Predictors and incidence of depression and anxiety in women undergoing infertility treatment: a cross-sectional study 

Dear Dr. Wang:

I'm pleased to inform you that your manuscript has been deemed suitable for publication in PLOS ONE. Congratulations! Your manuscript is now with our production department. 

Kind regards, 

on behalf of

MD Jacopo Di Giuseppe 

Academic Editor

PLOS ONE